# Queen honey bees exhibit variable resilience to temperature stress

**Alison McAfee** [1,2]*, **David R. Tarpy**[1], **Leonard J. Foster**[2]

**1** Department of Applied Ecology, North Carolina State University, Raleigh, North Carolina, United States of America, **2** Department of Biochemistry and Molecular Biology, Michael Smith Laboratories, University of British Columbia, Vancouver, British Columbia, Canada

\* amcafee@ncsu.edu

**Data Availability Statement:** All raw mass spectrometry data are available from the MassIVE repository (https://massive.ucsd.edu/ProteoSAFe/static/massive.jsp; accession number: MSV000086862). All other data are contained within the Supporting Information files.

## Abstract

Extreme temperature exposure can reduce stored sperm viability within queen honey bees; however, little is known about how thermal stress may directly impact queen performance or other maternal quality metrics. Here, in a blind field trial, we recorded laying pattern, queen mass, and average callow worker mass before and after exposing queens to a cold temperature (4°C, 2 h), hot temperature (42°C, 2 h), and hive temperature (33°C, control). We measured sperm viability at experiment termination, and investigated potential vertical effects of maternal temperature stress on embryos using proteomics. We found that cold stress, but not heat stress, reduced stored sperm viability; however, we found no significant effect of temperature stress on any other recorded metrics (queen mass, average callow worker mass, laying patterns, the egg proteome, and queen spermathecal fluid proteome). Previously determined candidate heat and cold stress biomarkers were not differentially expressed in stressed queens, indicating that these markers only have short-term post-stress diagnostic utility. Combined with variable sperm viability responses to temperature stress reported in different studies, these data also suggest that there is substantial variation in temperature tolerance, with respect to impacts on fertility, amongst queens. Future research should aim to quantify the variation and heritability of temperature tolerance, particularly heat, in different populations of queens in an effort to promote queen resilience.

## Introduction

The queen is normally the sole reproductive female within a honey bee colony and can live up to eight years, though normally not more than three [1]. Therefore, any impairment to her fecundity has a large impact on a colony's productivity and ability to withstand environmental challenges. Concerningly, poor queen quality is a frequently reported problem in beekeeping operations [2,3]. Indeed, survey results show that over half of colonies in some operations are requeened within the first six months of being established [4], but surprisingly little is known about why queens are failing so frequently [5].

When they are about one week old, queens embark on one or more nuptial flights and store a proportion of each mate's sperm, which must last for their lifetime since they are unable to mate again once they deplete their sperm stores. As such, a queen's lifetime reproductive

**Funding:** AM was supported by a fellowship from the Natural Sciences and Engineering Research Council (https://www.nserc-crsng.gc.ca/index_eng.asp). A Project Apis m. (https://www.projectapism.org/) grant to AM, LJF, and DRT funded the research, along with supporting grants from Genome Canada (264PRO) (https://www.genomecanada.ca/), Genome BC (https://www.genomebc.ca/), and the BC Ministry of Agriculture (https://www2.gov.bc.ca/gov/content/governments/organizational-structure/ministries-organizations/ministries/agriculture). The funders had no role in study design, data collection and analysis, decision to publish, or preparation of the manuscript.

**Competing interests:** The authors have declared that no competing interests exist.

capacity is ultimately sperm-limited, and any decrease in the viability of her stored sperm results in a permanent change to her fecundity and longevity [1,6]. Previous research has shown that extreme temperatures (both hot and cold) can reduce viability of stored sperm and fresh ejaculates, and temperature stress has been proposed as potential causal factor underlying queen failure in apicultural operations [7–10].

Queens are vulnerable to temperature stress during shipping and potentially inside small colonies during extreme weather events (*i.e.*, heatwaves) [7–9]. While some data suggests that temperature stress inside hives is theoretically possible [8,11], and extreme ambient temperatures are associated with colony losses [12], the actual risk that in-hive temperature fluctuations pose to queens is not known. This is because the core of colony is remarkably thermostable, whereas the periphery is the variable zone [8,11]–queens could therefore avoid temperature stress by remaining in the center of the brood nest. However, queen honey bees are routinely shipped both domestically and internationally to satisfy the seasonal needs of beekeepers, and they are unable to adequately thermoregulate during transit [7–9]. Canada imports over 200,000 queens annually, mainly from warmer geographic regions (predominantly the US, New Zealand, Australia, and Chile) to satisfy early season demand before locally produced queens become available [13]. In the United States, queens are mainly produced by a relatively small number of large-scale queen production operations in Hawaii, California, and the southeastern states for domestic distribution. Queens are typically shipped in small cages with few bees and poorly thermoregulated environments, leaving them vulnerable to perturbations in ambient temperatures, including in domestic shipments [7–9].

We previously determined that sperm viability does not change when queens are kept between 15 and 38°C; outside this range, stored sperm viability tends to decrease [8]. Interestingly, Withrow *et al.* [14] found that queens from honey bee packages experiencing deviant temperatures during transport were more likely to fail, despite those temperatures being well within the safe temperature zone. However, this effect was apparently not linked to sperm viability, suggesting that temperature may impact other aspects of queen physiology. Worryingly, extreme temperatures have been documented repeatedly during routine queen shipping [7–9]. While this is known to directly impact queen fertility via reductions in sperm viability, maternal effects of stress and vertical impacts on progeny are poorly characterized. In one study by Preston *et al.*, the researchers investigated vertical effects of cold stress (4°C, 2 h) applied to adult queens, and found that cold stress delayed development of embryos and adult emergence, but not adult immunocompetence nor behaviors [15]. Heat stress has not been investigated in this regard, and effects of temperature stress on egg laying pattern, queen mass, and vertical effects on global protein expression profiles have not been investigated.

Transgenerational effects of temperature stress have been described in other animals, including insects [16–19]. For example, eggs fertilized with sperm from heat-stressed males develop into adults with shorter lifespans and reduced reproductive potential in the flour beetle, *Tribolium castaneum* [16]. In a parasitoid wasp (*Aphidius ervi*), which is in the same Order (Hymenoptera) as honey bees, maternal heat stress increases the developmental time and decreases the survivorship of her progeny [17]. However, little is known about the physiological impacts of temperature stress on honey bee queens, including their performance within colonies and potential altered physiology of their progeny. Heat stress could therefore not only reduce the fertility of the queen, but also change the physiology of the eggs she lays and the resulting adults. As such, it is conceivable for there to be consequences of thermal stress on queens above and beyond that of her stored sperm, and indeed, some short-term impacts of cold stress have been identified [15].

Previously, we conducted short-term stress experiments (two hour exposures to extreme temperatures, followed by a two day recovery period) with caged queens to evaluate changes in

protein abundance in the spermathecal fluid, from which we proposed a panel of four candidate queen stress biomarkers linked to temperature stress [20]. Here, we conducted a blind field trial to more thoroughly investigate the effects of temperature stress on queen performance and productivity metrics. Reproductive potential (quality) can vary substantially among individual queens [21], so we recorded non-destructive quality metrics before and after exposure to thermal stress in order to account for changes relative to an individual queen's baseline. We also performed proteomic analysis on eggs laid before and after the queen was stressed to investigate potential vertical impacts of temperature stress (*e.g.*, through differential egg provisioning or embryo development). At the end of the experiment, we sacrificed queens to measure their stored sperm viabilities and abundance of candidate stress biomarkers to evaluate their utility for field-realistic stress diagnostics.

## Results and discussion

### Cold stress reduces sperm viability, but laying pattern and queen mass are unaffected by heat and cold stress

To determine how queen performance is impacted by temperature stress, we performed a blind field trial in which we compared queen laying pattern and queen mass before and after exposing the queen to extreme cold (4°C, 2 h, n = 9), extreme heat (42°C, 2 h, n = 10), and a handling control (33°C, 2 h, n = 9). This experimental design enabled us to account for phenotypic differences among individual queens, which had different baseline body weights and laying patterns. We chose these temperatures and exposure durations because they have been previously shown to significantly affect sperm viability [7], yet are short enough durations that they still have a reasonable chance of occurring during shipping. We found no effect of heat stress on any of these phenotypes (Fig 1A and 1B) indicating that, contrary to our expectations,

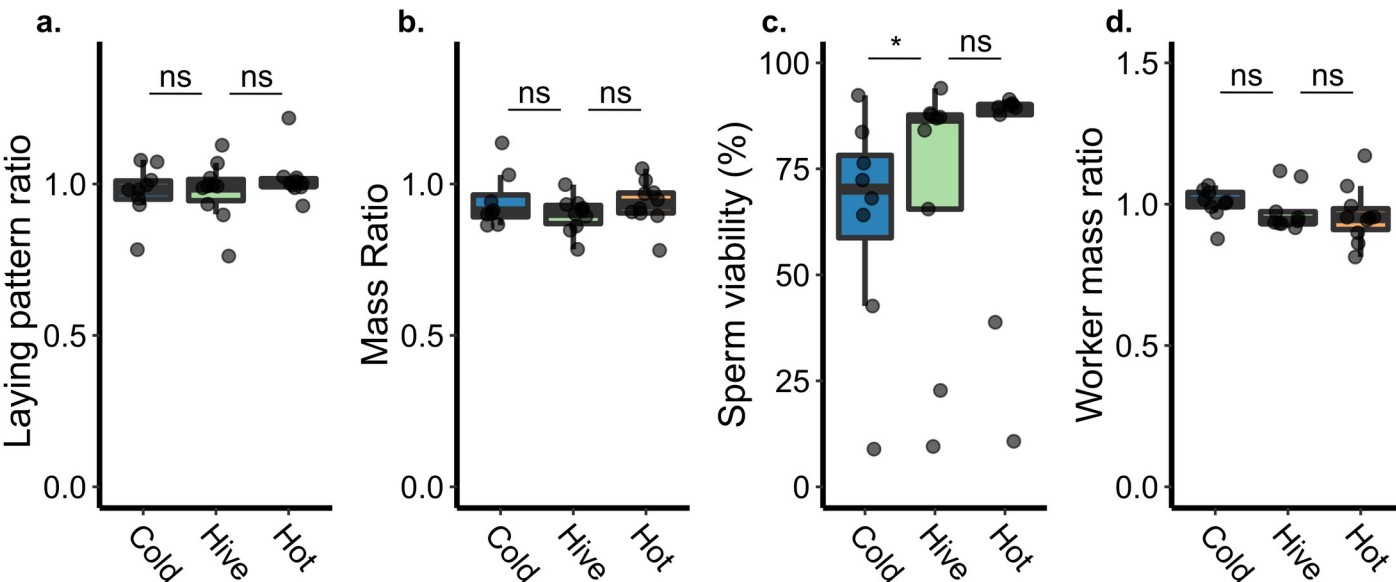

**Fig 1. Queen phenotypes associated with temperature stress.** See Table 1 for complete statistical parameters. Non-significant = ns. Boxes represent the interquartile range, bars indicate the median, and whiskers span 1.5 times the interquartile range. Sample sizes shown in panel a. apply to all. All ratios represent post-stress relative to pre-stress. We retrieved locally produced queens from their established colonies and exposed them to extreme cold (4°C, 2 h, n = 9), extreme heat (42°C, 2 h, n = 10), or a handling control (33°C, 2 h, n = 9). We recorded phenotypic metrics a) egg laying pattern and b) queen mass before and two weeks after applying the stress. c) Sperm viability is a destructive measurement and was therefore recorded only after the stress. d) Callow worker mass was determined by sampling between 9 and 12 callow workers from each colony and calculating the average mass. In all cases, ratios refer to post-treatment measurements relative to pre-treatment.

the queens themselves are apparently resilient to heat stress. Cold stress, however, did significantly reduce sperm viability (generalized linear model, binomial distribution with logit link function; df = 23, z = -1.97, p = 0.048), but laying pattern and queen mass was unaffected. Although initially surprising that cold stress could reduce sperm viability but not affect egg laying pattern, it could simply be that although sperm viability decreased, each egg still became fertilized in this short-term experiment since queens release multiple sperm per egg [6] and these were young queens which had not yet substantially decreased their sperm stores. Statistical parameters for all tests, including the model used, degrees of freedom, t or z statistics, and p values, are summarized in Table 1.

The effect sizes (Cohen's d) for all contrasts were low, ranging from d = 0.03 (laying pattern ratios of cold-treated queens compared to controls) to d = 0.62 (the queen mass ratios of cold-treated queens compared to controls) (Table 1), indicating that, assuming a difference between groups does exist, a sample size of n = 33 to 13,740 would be needed in each group to achieve a power of at least 0.80 for all parameters. While a sample size of 33 would be feasible, indicating that perhaps some differences were missed due to insufficient sample sizes, for many of the parameters measured, the corresponding sample size needed is unreasonably high. As a reductionist approach, the average effect size across all our comparisons of interest was 0.34, which would necessitate a sample size of 108 colonies in each group (324 in total), which is not a realistic scale. Rather, the typically small effect sizes indicate that the strength of phenotypic responses to temperature stress is generally low.

We and others have previously found that exposure to hot and cold temperatures significantly reduces viability of stored sperm [7–9], as well as drone ejaculates [8,10]. While cold stress did reduce sperm viability, here we detected no effect of heat, despite conducting similar stress treatments, in terms of temperature, duration, and sample size, to what has been previously published (Fig 1C) [7–9]. There are several potential explanations for this null result, none of which are necessarily mutually exclusive: 1) previous studies were not conducted blindly, which could have adversely influenced the previously published differences; 2) our sperm samples could be contaminated with venom, obscuring an effect of temperature; 3) not all queens are equally sensitive to temperature stress, and these queens were more resilient than those in previous studies; and 4) not all sperm are equally sensitive to temperature stress, and the sperm stored in these queens were particularly resilient.

The impact of blinding on experimental results is well known and has been described with respect to insect sperm viability specifically [22]. Worryingly, Eckel *et al.* found that, using the common bed bug (*Cimex lectularius*) as a test system, when experimenters knew the hypothesis being tested, they identified the expected significant differences in sperm viability among groups and the null hypothesis was rejected. However, when the experiment was repeated by a blind experimenter, they found no significant differences. Blinding is still an uncommon

**Table 1. Statistical parameters for queen and worker quality metrics.**

| Independent variable | Test | df | Contrast | Effect size | \|t/z\| | P |
|---|---|---|---|---|---|---|
| Sperm viability | Generalized linear model (glm) | 23 | Hot:Control | 0.20 | 1.12 | 0.26 |
| | | | Cold:Control | 0.22 | 1.97 | 0.048* |
| Pattern ratio | Linear model (lm) | 27 | Hot:Control | 0.48 | 1.01 | 0.32 |
| | | | Cold:Control | 0.03 | 0.06 | 0.95 |
| Queen mass ratio | Linear model (lm) | 25 | Hot:Control | 0.58 | 1.10 | 0.28 |
| | | | Cold:Control | 0.62 | 1.28 | 0.21 |
| Average worker mass ratio | Linear model (lm) | 26 | Hot:Control | 0.20 | 0.45 | 0.65 |
| | | | Cold:Control | 0.40 | 0.67 | 0.51 |

**Table 2. Spermathecal fluid proteins significantly correlating with sperm viability[a].**

| Protein[b] | Name | Log(fold change) | t | P value | Adjusted p value |
|---|---|---|---|---|---|
| XP_006572091.1 | restin homolog | -0.067 | -5.67 | 4.47E-06 | 0.0077 |
| XP_026297189.1 | A disintegrin and metalloproteinase with thrombospondin motifs 12 | -0.075 | -5.80 | 5.54E-06 | 0.0077 |
| XP_006560640.1 | glutamyl aminopeptidase-like | -0.078 | -6.11 | 9.02E-06 | 0.0084 |
| XP_016768886.2 | xanthine dehydrogenase | 0.054 | 5.44 | 2.16E-05 | 0.015 |
| XP_006560925.2; XP_393531.4; XP_006560924.2; XP_016766611.2; XP_006560926.2 | serine/threonine-protein phosphatase 4 regulatory subunit 4 isoform X3 | -0.034 | -5.92 | 3.77E-05 | 0.021 |
| NP_001011607.1 | melittin precursor | -0.066 | -5.47 | 8.33E-05 | 0.039 |

[a]Proteins significant at 5% FDR are shown.

[b]Multiple accessions indicates that these proteins were indistinguishable based on the peptides identified.

practice in biological sciences, and to our knowledge previous studies on honey bee sperm viability and temperature stress have not been conducted blindly, which may have influenced previously published results.

We initially suspected that venom contamination could have obscured our results, because upon inspection of proteins quantified within the spermathecal fluid, we found that several proteins significantly correlating with sperm viability (5% false discovery rate), one of which was melittin precursor (hereon referred to as melittin; Table 2). While the relationship among the other proteins and sperm viability is unclear (restin homolog, for example, is associated with the cytoskeleton filaments and Hodgkin's disease in mammals, but the most similar *Drosophila* sequence is the uncharacterized protein Dmel_CG14354), melittin is the main protein component of honey bee venom and is negatively correlated with sperm viability, which would be consistent with contamination causing sperm death. However, melittin transcripts have been previously identified in the spermathecae of mated queens by an independent research group [23] and another group found melittin proteins in both virgin and mated queen spermathecal fluid [24]. We searched for melittin in our own previously published proteomics data (three independent datasets) and found that it was consistently present [8,20,25]. It is unlikely that all of these previously conducted studies suffered from venom contamination; rather, melittin is likely an endogenous protein to the spermatheca. Indeed, melittin was also previously identified in drone ejaculates, and drones do not even possess venom glands; therefore, this protein is likely multifunctional with an unknown role in reproduction or other biological processes. Furthermore, we identified melittin in 103 out of 123 samples in a recent study in which we also analyzed sperm viability, where we found that melittin held no significant relationship with this metric [25]. While the presence of endogenous melittin in the spermatheca is not necessarily mutually exclusive with adverse effects of venom contamination, the apparent natural occurrence of venom proteins in the spermatheca makes contamination seem unlikely. In the present work, correlations between melittin as well as other significant proteins with sperm viability (which were not identified in previous studies) suggests that spermathecal proteins associated with sperm viability may not be constant, and we speculate that different proteins may be important for sperm maintenance at different life stages, for example.

Variation in thermal tolerance of individual queens and the sperm they contain has not been explicitly studied, and these factors may be influencing our results. Indeed, although previously published studies all identified a significant effect of temperature, the magnitude of the impact on sperm viability varies considerably. For example, Rousseau *et al.* observed a significant sperm viability drop of 14% when queens were exposed to 40˚C for two hours [9]. Pettis

*et al.*, however, found that exposures to 4˚C and 40˚C for two hours reduced sperm viability from ca. 65% down to ca. 22% [7]. Yet, in a subsequent study conducted by the same researcher, results showed that two hours at 40˚C was not sufficient to significantly reduce sperm viability. In that study, a significant relationship between temperature and sperm viability was only apparent either when contrasting the control with the most extreme exposure tested (42˚C for four hours), or when considering all data across a range of temperatures collectively, [8]. Our data are therefore not necessarily contradictory with previous work, which is highly variable and has not always identified significant differences associated with temperature when considering singular contrasts [8]. Despite being fundamental to understanding effects of temperature stress on queens, this variation in queens' responses has not previously been explicitly discussed.

Finally, it is also possible that variation of thermal tolerance of the sperm themselves (*i.e.* the "fragile sperm hypothesis"; see [10]) could be causing these discrepant results. It is plausible that not all sperm are the same; indeed, not all drones are the same (reviewed by [26])— they vary significantly in their reproductive quality [27], colony environment (Metz *et al.*, unpublished data), and their thermal tolerance [8,10]. For example, heat stress survival curves of age-matched drones show that after six hours of 42˚C exposure, half of drones survived, while others began to perish after just three hours [8]. It stands to reason that if drones of variable quality may still inseminate a queen, how well their respective sperm tolerate stressful conditions may similarly vary. As such, the queens in the present study may have happened to mate with drones that have robust sperm, whereas those in other studies may have mated with drones whose sperm were perhaps less tolerant to stress. Such tolerance could further explain why we did not see the expected effect of temperature, and warrants further investigation.

## No vertical effects of queen stress on callow worker mass nor egg proteins

Transgenerational effects of abiotic stressors have been documented in other insects, including hymenoptera [16,17,19], but they have only been briefly investigated in honey bees with regard to cold stress, and not heat stress [15]. In a previous study, we identified proteomic changes in the ovaries of queens shortly (2 d) after being temperature stressed [8], indicating a potential route for a maternal effect of temperature stress. To test the hypothesis that abiotic stress on the queen could affect her progeny, we sampled callow workers (n = 9–12 per colony) that developed from eggs the queen laid before and after being stressed and recorded their emergence mass. We found no differences in worker mass ratios (post-stress relative to pre-stress) between treatment groups (Fig 1D; see Table 1 for statistical parameters).

Despite there being no effect on adult mass at emergence, there could have been more subtle developmental changes early in life; for example, differences in egg provisioning could lead to altered embryo development. We sampled 48–72 h old eggs (n = ~30 eggs per sample) laid by queens before and after treatment and quantified proteins by label-free quantitative proteomics to determine if temperature stress impacts the proteins expressed in or provisioned to eggs. We identified 4,604 unique protein groups at a 1% false discovery rate, which is the richest honey bee proteomics dataset yet reported. There is one other report with higher numbers (8,609 proteins) [28], but the authors' definition of a protein was non-parsimonious (*i.e.*, all proteins within a protein group were counted as unique identifications) thereby inflating the results in a way that is not the accepted standard in the field. Despite these rich data, we still did not identify any protein expression differences among treatment groups even at a relatively loose FDR (multiple samples test (Perseus), Benjamini-Hochberg correction, 10% FDR), indicating that there are no vertical effects of queen temperature stress on embryo protein expression (Fig 2A).

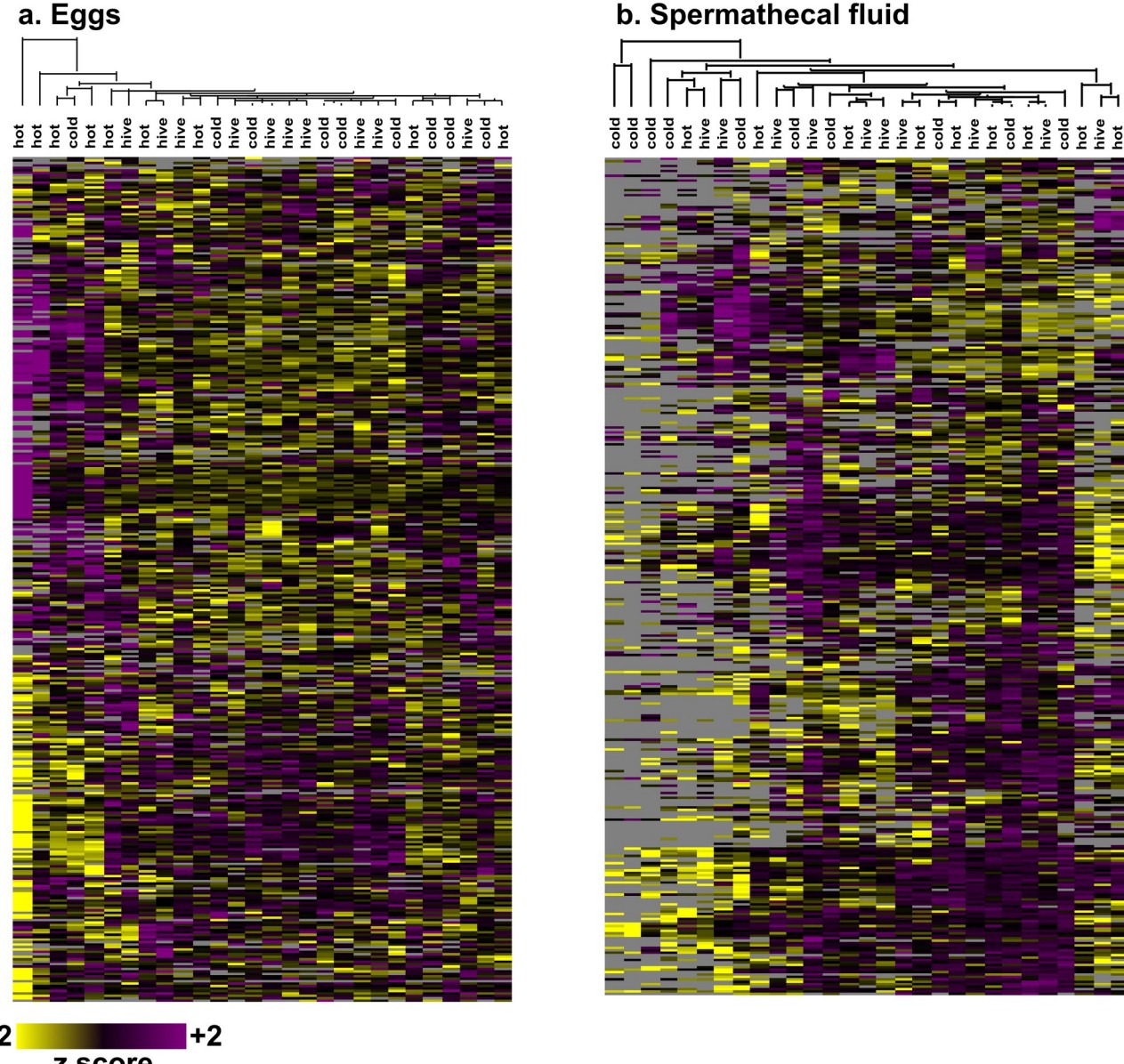

**Fig 2. Proteomics analysis of eggs and queen spermathecal fluid.** We used mass spectrometry-based quantitative proteomics to evaluate protein abundances in mature (>48 h old) eggs produced by queens before and after exposure to extreme temperatures ('cold' = 2 h at 4°C, 'hot' = 2 h at 42°C, and 'hive' control temperature = 2 h at 35°C) (a) and spermathecal fluid of temperature-stressed (b) queens. Egg data is expressed as the protein LFQ intensity ratio of eggs laid after stress relative to before stress. All data was z-scored prior to plotting. We identified 4,604 and 3,127 proteins in eggs and spermathecal fluid, respectively, using MaxQuant (1% FDR). No statistically significant differences were identified between experimental groups (limma, Benjamini-Hochberg correction to 10% FDR). Heatmaps were generated using Perseus v1.6.1.1 (clustered via Euclidian distance, 300 clusters, 10 iterations). Both columns and rows were clustered, but only column dendrograms are shown for clarity.

These data show no obvious effect of maternal temperature stress on the eggs (embryos) or adult phenotypes, which does not support the hypothesis that maternal temperature stress adversely impacts the queen's progeny. We acknowledge, however, that we did not record an exhaustive list of all potential phenotypes (*e.g.*, behavioral ontogeny), and it is possible that there is some impact on progeny that is as yet unrealized. Regardless, the data that we acquired also show a lack of apparent impact on the queen's phenotype (body mass) and productivity

(laying pattern and sperm viability), which point to a surprisingly robust stress-tolerance system. Indeed, queens are highly tolerant of heat stress, in terms of survival, compared to drones [8]. Since similar reductions in sperm viability are observed when both ejaculates and queens are heat-shocked [8], even the negative effects of heat on sperm stored within the queen can be explained by direct impacts on the sperm cells rather than indirectly through heat harming the queen's ability to keep the sperm alive. It is also possible that the effect of temperature stress is impacted by extraneous co-exposures to other variables, like pathogens, nutritional stress, or pesticides, which may not always be accounted for.

Evidence consistently points to queens having a high tolerance to temperature stress, in terms of survival, which is puzzling because queens do not often naturally experience extreme temperatures; therefore, there should be little opportunity for selective pressure to drive the evolution of extreme temperature tolerance. It is possible that the temperature variation experienced during mating flights and swarming is sufficient to select for temperature tolerance. However, drones, which similarly execute mating flights, are mortally sensitive to temperature extremes [8,10], among other stressors [29–32]. This is likely at least in part due to their haploid genome—without the potential for compensatory alleles, individuals with recessive deleterious or "susceptible" genotypes are more likely to manifest and are thus purged from the population [33,34].

It has been previously proposed that this "haploid susceptibility," where haploid males more sensitive to stressors than their diploid counterparts, has an adaptive advantage: culling harmful mutations or unfavourable recessive alleles from the gene pool [33–35]. Although it is possible that queens' exposure to temperature fluctuations during mating flights and swarming is theoretically sufficient to select for a wide range of tolerance, we think that is an unlikely scenario. Rather, we speculate that selection for queen temperature tolerance is at least in part achieved indirectly through the drones, which share 100% of their genetic material with their mother queen, contribute to 50% of the genetic material to their daughters (which may become queens), exhibit well-documented susceptibility to temperature and other stressors [8,10,29–32], and are also more likely to be exposed to extreme temperatures since they themselves engage in more mating flights and occupy the periphery of the nest, which is relatively poorly thermoregulated [36].

As such, drones can be thought of as physical extensions of the queen, providing bodies more exposed to selective pressure, ultimately helping to drive selection for robust stress tolerance in queens and females in general. We admit, however, that this does not entirely explain queens' resilience to extreme temperatures. Given that 50% to 77% of drones die after several hours of heat exposure, and that drones share 100% of their genes with their queen, we would expect at least some queens to have the unfortunate genetic combinations leading to heat sensitivity, and therefore other factors must also contribute.

## Longevity of previously identified queen stress biomarkers is less than two weeks

Previously, we developed a panel of candidate stress protein biomarkers to aid with queen failure diagnostics [20]. Among others, this panel consists of four proteins that are uniquely upregulated in spermathecal fluid of queens exposed to heat-stress and cold-stress relative to controls [20]. The length of time that these markers remain upregulated in the spermathecal fluid directly impacts their utility for queen failure diagnostics because symptoms of queen failure may not manifest or be noticed until long after a queen stress event has occurred. Our initial experiments identifying the candidate stress markers evaluated queens two days after conducting the acute (2 h) temperature exposure [20], but after a longer period of time marker

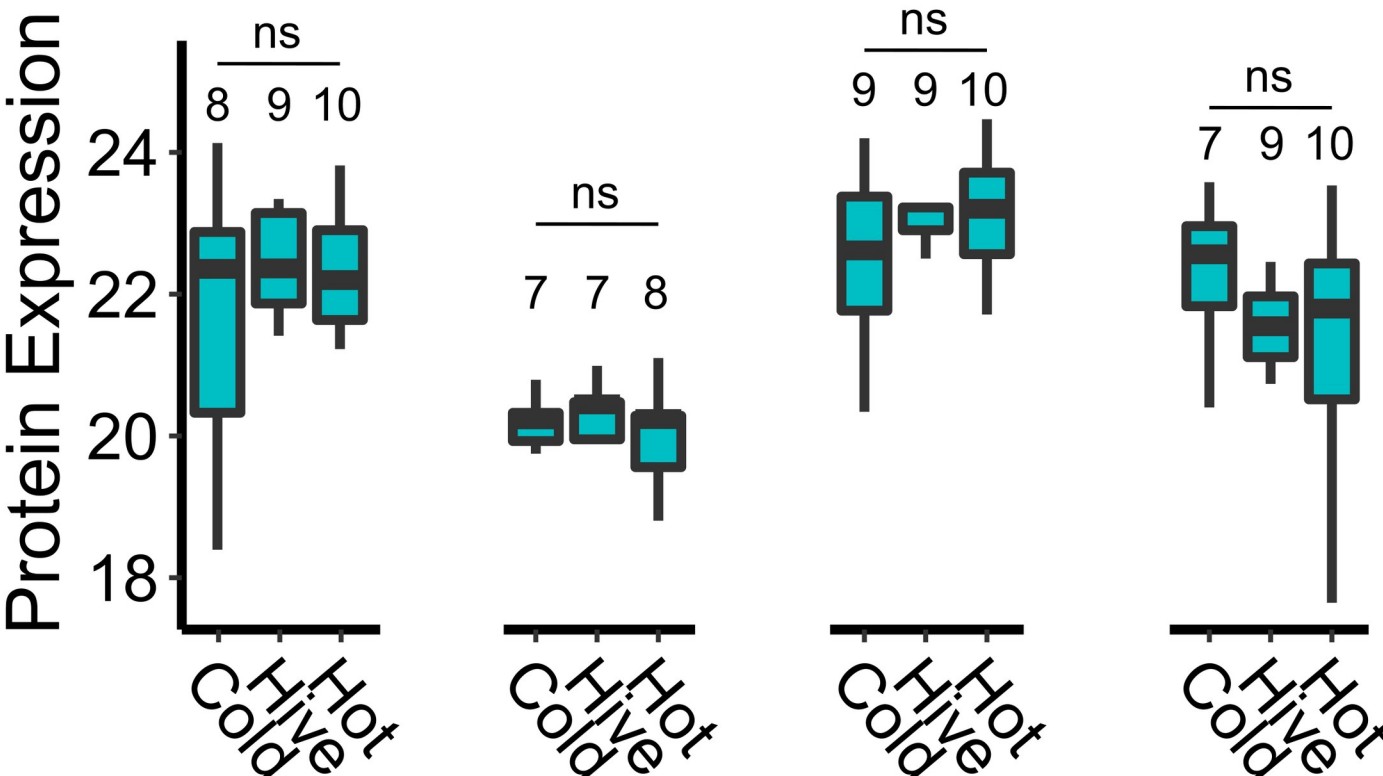

**Fig 3. Candidate queen stress biomarker expression in spermathecal fluid.** We extracted protein expression (log2 transformed normalized LFQ intensities) data for each of our previously defined candidate queen stress biomarkers and compared expression between exposure and control groups (Dunnett's test, control = 'Hive' temperature). ns = non-significant. Boxes represent the interquartile range, bars (where box sizes permit) indicate the median, and whiskers span 1.5 times the interquartile range. Sample sizes are numbered on the figure (there is variation between proteins due to missing values). No significant differences were identified ($p > 0.05$ for all comparisons).

abundance may fade or the stress response may evolve to a different protein fingerprint. Here, we evaluated global changes in spermathecal protein abundances, as well as the candidate stress markers specifically, in queens two weeks after temperature stress exposure to identify potential longer-term candidate markers and to evaluate the temporal stability of the ones we already proposed. Unexpectedly, we found that no global changes in protein expression were present two weeks after the queens were exposed to stressors (Fig 2B; limma, Benjamini-Hochberg false discovery correction to 10% FDR).

Despite the lack of globally differentially expressed proteins, we still checked our original candidate temperature stress biomarkers specifically, since evaluating four specific proteins does not suffer from substantially reduced statistical power owing to reduced multiple hypothesis testing. We also did not identify differences in expression of these *a priori* defined proteins of interest (XP_026296654.1, XP_395122.1, XP_001120006.2, and XP_395659.1) (Fig 3), indicating that either (a) two weeks is sufficient time for them to revert to constitutive expression levels and that they are mainly an acute stress response mechanism, or (b) the stress treatments did not elevate these biomarkers in the first place. However, we note that candidate heat-stress

biomarker expression is a reproducible response in queens from different genetic sources (Californian and multi-generational local queens) [8,20]. Furthermore, the candidate markers XP_001120006.2, and XP_395659.1 are also conserved heat-shock proteins (HSPs); therefore, it is highly unlikely that these proteins simply failed to become upregulated at all after heat-shock. Replication of the cold-shock markers has not yet been attempted, but regardless of whether this stressor induces different proteins in different queens or if the proteins were induced but regressed back to baseline expression, these data show that protein markers expressed shortly after queen stress, as previously determined, do not have sufficient stability for routine diagnostic purposes. Coupled with a general lack of observable effects on other queen performance metrics and no apparent vertical impacts of queen stress on workers, these results are an encouraging tribute to queens' potential for resilience against temperature stress.

We caution that although no significant effects were identified in these experiments, with the exception of cold stress reducing sperm viability, this does not mean that all queens are universally resilient. Indeed, an abundance of previous work clearly shows that heat stress can seriously impact queen quality by damaging their sperm as well [7–9], and it is possible that our field observations did not extend long enough to observe additional potential negative, vertical effects of queen stress. Although there was no molecular evidence of queen stress at the end of the experiment, we only analyzed the queen's spermatheca and her eggs, not other tissues, which could also have been affected.

## Conclusion

Queen failure is a common problem in beekeeping operations, but surprisingly little is known about the underlying factors. Here, we investigated short-term (two weeks post-stress) impacts of heat-stress and cold-stress on queen quality metrics. We confirmed that cold stress reduces stored sperm viability, but surprisingly, this could not be confirmed for heat stress. Furthermore, we found no significant effects of temperature stress on laying pattern, queen mass, or average callow worker mass, nor protein expression differences in eggs laid before and after stress exposure. We also identified no molecular evidence of the stress event in the queens' spermathecal fluid, indicating that two weeks post-stress may be sufficient time for previously identified candidate stress biomarkers to revert back to baseline expression. Some of these observations are contrary to previous findings and suggest that there is variation in queens' abilities to tolerate extreme temperatures. Cumulatively, the data acquired consistently demonstrate resilience of the queen herself to temperature stress, even if it may decrease the viability of the sperm she stores in some cases. Future research should focus on longer-term experiments to test queen quality after stress throughout a beekeeping season, as well as investigate the drivers of variation in sperm viability responsiveness that has been reported in the literature. Extreme temperature fluctuations should also be investigated (*e.g.*, from extreme cold to extreme heat) or longer durations of milder temperatures, which may be realistic scenarios to occur as the climate continues to change.

## Materials and methods

### Honey bee colonies

As a non-cephalopod invertebrate species, honey bees are not subject to animal ethics committee approval at the University of British Columbia.

We purchased twelve 1.5 kg honey bee packages from Tasmania in order to ensure that our experimental honey bees were all free from mites at the beginning of the experiment. Packages were installed in standard 10 frame deep hive bodies and supplied with pollen and syrup to encourage brood rearing and population growth. After 1.5 months (two brood cycles), we split

the colonies into thirty 3-frame nucleus colonies (nucs; 1 frame of honey, 1 frame with open brood covered in bees, 1 frame with capped brood) and supplied each nuc with a caged, mated queen from New Zealand, a frame feeder for light syrup (~35% sucrose) and a ½ lb (ca. 227 g) pollen patty (15% protein), which we fed continuously throughout the duration of the field trials except for during a period of nectar flow and population growth when we replaced the frame feeders each with two frames of drawn comb. All nucs were kept in a single apiary in Richmond, Canada. The New Zealand queens were allowed to head the colonies for several weeks until the beginning of the experiment (June), at which time they were removed and replaced with local queen cells produced as described below. The day that queen cells were introduced, any emergency queen cells were destroyed.

## Temperature stress field trial

To eliminate a potential confounding effect of extreme temperature exposure during shipping, we reared queens used for the temperature stress trial locally. We produced queen cells using standard queen rearing techniques (1–2 d old larvae were grafted from a single colony into queen cups and inserted to a queenless cell builder) [37]. As putative half-sisters, these queens had some genetic differences, but this approach offers the highest level of genetic control apart from grafting from single-drone inseminated queens. Twelve days after grafting, we supplied each experimental nuc with a capped queen cell. Three days after adding the queen cell, queen cups were checked for successful emergence, and two weeks after queen cell introduction, we checked the nucs to see if the queens were laying eggs. Five queens were either laying poorly (few eggs or multiple eggs per cell) or were not laying, and these were replaced with queens which had been concurrently reared and mated in a second nearby apiary. At this time, we also re-normalized colony populations by adding 1 or 2 cups of nurse bees, as needed, from the strongest colonies to the weaker colonies. To do this, the queens were first located, then nurse bees were separated from foragers by shaking frames of bees into a large plastic bin, and shaking the bin for several minutes to agitate foragers to fly away, leaving only the non-flying young Two colonies became queenless during the course of the experiment (in the control group and one in the cold group), leaving 28 queens at the time of experiment termination.

Once each queen had been laying eggs for at least two weeks, we evaluated their laying patterns by locating a patch of approximately 100 eggs and recording how many cells within that patch were missed (*i.e.*, the cell was not otherwise occupied but lacked an egg). We avoided patches at the edge of the brood area. If a patch included an occasional cell with a newly eclosed larva, it was counted as 'laid' since it was likely that the eggs were simply on the verge of hatching. This method does not distinguish between eggs that were laid and then cannibalized by workers; however, since all colonies were fed supplemental protein, egg cannibalization should be linked to developmental deficiencies rather than nutritional stress, which is a desirable feature to which our method should be sensitive. We repeated this procedure two weeks after the queens were experimentally stressed in order to calculate a change in laying pattern (the ratio of the fraction of cells laid post-stress relative to pre-stress).

On the day that laying pattern was evaluated, queens were caged in plastic JZ-BZ cages with five attendants and candy, then transported to the laboratory where a colleague not otherwise involved in the study briefly anesthetized the queens with carbon dioxide (5 min), weighed them on an analytical balance, and randomized them into three treatment groups (n = 9: cold-stress at 4°C for 2 h, n = 10: heat-stress at 42°C for 2 h, and n = 9: control at 33°C for 2 h), keeping the experimenter blind to their assignments. Humidity was supplied by a water pan placed in each incubator, and all queens experienced stress treatments in the dark (however, they were exposed to light during transport and weighing). Queens were then transported

back to the apiary and re-introduced to their respective colonies, but remained caged. After two days, we sampled approximately 30 pooled eggs per colony (at which time all eggs were >2 and <3 days old), froze them on dry ice, and released the queens. Two weeks post-stress, laying patterns were again evaluated, the queens were transported to the laboratory, anesthetized, weighed, and sacrificed for sperm viability analysis. Spermathecae were removed from the abdomen with forceps and blotted dry on a kimwipe, then clean forceps were used to gently remove the tracheal net surrounding the spermatheca. The spermatheca was then lysed in an Ependorf tube containing Buffer D (17 mM D-glucose, 54 mM KCl, 25 mM NaHCO$_3$, 83 mM Na$_3$C$_6$H$_5$O$_7$) and sperm viability was measured using dual fluorescent staining exactly as previously described [25]. Two days after queens were sacrificed, we returned to the nucs to sample remaining eggs as already described and to introduce a new, commercially supplied queen to maintain the colony in a queenright state until post-stress callow workers could be collected.

We collected newly emerged (callow) workers from each colony four weeks after the beginning of the experiment and four weeks after the queens were stressed. It takes 21 days for worker eggs to develop into adults; therefore, callow workers collected four weeks after the beginning of the experiment developed from eggs laid one week after the experiment began (*i.e.*, before the queens were stressed). Likewise, callow workers collected four weeks after the queens were stressed developed from eggs laid one week post-stress. Callow workers are easily recognizable due to their light grey color, soft bodies, and inability to fly. We collected 9–12 workers per colony per time point in order to calculate a change in average mass at emergence.

## Proteomics analysis

Proteins were extracted, digested, and purified from spermathecal fluid exactly as previously described [20]. Briefly, sperm cells were spun down from the Buffer D-diluted spermathecal fluid solution and soluble proteins in the supernatant were precipitated with four volumes of ice-cold acetone. The pellets were washed and resuspended in urea digestion buffer (6 M urea, 2 M thiourea, in 100 mM Tris, pH 8). Preliminary tests showed that each spermatheca yields approximately 5–10 µg of protein. The proteins were reduced, alkylated, then digested with 0.2 µg of Lys-C (3 h, room temperature) followed by 0.2 µg of trypsin (overnight, room temperature). Peptides were desalted using in-house made C18 STAGE-tips, dried, suspended in Buffer A (0.1% formic acid, 2% acetonitrile), and quantified using a Nanodrop (280 nm absorbance). One µg of peptides were injected on a Thermo easy-nLC 1000 liquid chromatography system coupled to a Bruker Impact II mass spectrometer. Sample orders were randomized prior to loading, and instrument parameters were set exactly as previously described [20]. We followed the same procedure for analysis of the egg proteins, except that protein was extracted into 6 M guanidinium chloride (in 100 mM Tris, pH 8) using a Precellys homogenizer with ceramic beads. We digested approximately 25 µg of protein per sample using 0.5 µg of Lys-C and trypsin.

Raw mass spectrometry data were searched using MaxQuant (v 1.6.1.0) exactly as previously described [20]. We used the most recent honey bee canonical protein database available on NCBI (HAv3.1, downloaded November 18[th], 2019) with honey bee pathogen sequences added. Protein and peptide identifications were filtered to 1% FDR based on the reverse hits approach. All specific search parameters are available within the mqpar.xml file included in our data repository (see Data Availability).

## Statistical analysis

We analyzed sperm viability, laying pattern ratio, queen mass ratio, and worker mass ratio data using R (v3.5.1). For sperm viability analysis, the data was non-normal; therefore, we used

a generalized linear model (glm(), family = binomial) with live sperm counts as 'successes' and dead sperm counts as 'failures,' as has been conducted previously [38]. Ratio data were analyzed using a linear mixed model. All queens were reared locally but came from two different genetic lineages and mating locations, which is noted in the sample metadata provided (**S8 Table**); however, we did not include queen source as a random effect because this variable has only two levels, which can lead to inaccurate estimation of variance. All other response variables were analyzed using a linear model with Gaussian distribution.

For spermathecal fluid data, protein intensities ('LFQ intensity' columns from the Max-Quant output) were first log2 transformed, then reverse hits, contaminants, protein groups only identified by site, and protein groups without at least three defined values per treatment group were removed, leaving 1,899 proteins quantified. Differential expression analysis was performed using limma() (example code is provided, see Data Availability) and a Benjamini-Hochberg multiple hypothesis testing correction. We analyzed expression of individual candidate biomarkers (four proteins) using a Dunnett's test (hive temperature = control). For egg proteomics data, we calculated the ratio of each protein's expression post-stress relative to pre-stress prior to log2 transformation; otherwise, analysis was conducted using limma() following the same procedure as spermathecal fluid data. After filtering, 4,184 egg proteins were quantified. Heatmaps were generated using Perseus v1.6.1.1 (clustered via Euclidian distance, 300 clusters, 10 iterations).

## Data availability

Raw mass spectrometry data for the spermathecal fluid and egg analyses are available on the MassIVE proteomics archive (massive.ucsd.edu; accession MSV000086862). We have also made the protein group tables and experimental design tables more easily accessible as supplementary information (see S1 Table for spermatheca metadata, S2 Table for spermatheca proteomics data, S3 Table for limma results linking spermathecal protein expression to sperm viability, S4 Table for candidate biomarker data, S5 Table for egg sample metadata, S6 Table for egg proteomics ratios, S7 Table for egg proteomics intensities, and S8 Table for all phenotypic data associated with this field experiment. R scripts underlying data analysis and figure generation are available in S1 File (limma analysis) and S2 File (biomarker expression comparison).

## Supporting information

**S1 Table. Spermathecal fluid sample metadata.**
(XLSX)

**S2 Table. Spermathecal fluid proteomics data.**
(XLSX)

**S3 Table. Spermathecal fluid limma results for sperm viability factor.**
(XLSX)

**S4 Table. Spermathecal fluid candidate biomarker data.**
(XLSX)

**S5 Table. Egg sample metadata.**
(XLSX)

**S6 Table. Egg proteomics data ratios.**
(XLSX)

**S7 Table. Egg proteomics data LFQ intensities.**
(XLSX)

**S8 Table. Field phenotypic data for each colony and queen.**
(XLSX)

**S1 File. Example R code for limma analysis.**
(TXT)

**S2 File. Example R code for biomarker analysis.**
(TXT)

## Acknowledgments

We would like to acknowledge Julia Common for her help with queen rearing, Abigail Chapman for enabling blind treatments, and Jordan Tam and Bradford Vinson for assistance in the field.

## Author Contributions

**Conceptualization:** Alison McAfee, David R. Tarpy.

**Data curation:** Alison McAfee.

**Formal analysis:** Alison McAfee.

**Funding acquisition:** Alison McAfee, David R. Tarpy, Leonard J. Foster.

**Investigation:** Alison McAfee.

**Resources:** Leonard J. Foster.

**Supervision:** David R. Tarpy, Leonard J. Foster.

**Visualization:** Alison McAfee.

**Writing – original draft:** Alison McAfee.

**Writing – review & editing:** David R. Tarpy, Leonard J. Foster.

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
