## [Decision Letter · Decision Letter 0]

15 Jun 2021

PONE-D-21-11704

Queens exhibit variable resilience to temperature stress

PLOS ONE

Dear Dr. McAfee,

Thank you for submitting your manuscript to PLOS ONE. After careful consideration, we feel that it has merit but does not fully meet PLOS ONE’s publication criteria as it currently stands. Therefore, we invite you to submit a revised version of the manuscript that addresses the points raised during the review process.

Both referees and I thought that the study is interesting and that negative results do not preclude publication. The methods are well designed

However, several points should be considered in a revision, linked to analyses and other aspects:

- consider performing a power analysis to evaluate what size effect you could have reached with this dataset

- The applied context of transport  is valid, however your study may have other implications, ans as referee 2 points out, queens will go on several mating flights and encounter lower temperature for period ot time similar to the experimental one. Thus the control used have to be renamed adequately.

- improve figures

- Consider the possibility of using a dataset external to the Lab to pinpoint the contamination origin (or justify your hypothesis on venom contamination)

- evaluate the necessity of hove origin as a random effect

Both referees also have other minor comments which should be adressed in the rebuttal letter.

We look forward to receiving your revised manuscript.

Kind regards,

Nicolas Chaline

Academic Editor

PLOS ONE

Journal Requirements:

1. Please ensure that your manuscript meets PLOS ONE's style requirements, including those for file naming. The PLOS ONE style templates can be found athttps://journals.plos.org/plosone/s/file?id=wjVg/PLOSOne_formatting_sample_main_body.pdf and https://journals.plos.org/plosone/s/file?id=ba62/PLOSOne_formatting_sample_title_authors_affiliations.pdf

2. During your revisions, please note that a simple title correction is required: in order to provide clarity for readers, we would ask that you amend the title to "Queen honey bees exhibit variable resilience to temperature stress". Please ensure this is updated in the manuscript file and the online submission information.

Additional Editor Comments (if provided):

Reviewers' comments:

Reviewer's Responses to Questions

**Comments to the Author**

1. Is the manuscript technically sound, and do the data support the conclusions?

Reviewer #1: Yes

Reviewer #2: Yes

2. Has the statistical analysis been performed appropriately and rigorously? 

Reviewer #1: Yes

Reviewer #2: Yes

3. Have the authors made all data underlying the findings in their manuscript fully available?

Reviewer #1: Yes

Reviewer #2: Yes

4. Is the manuscript presented in an intelligible fashion and written in standard English?

Reviewer #1: Yes

Reviewer #2: Yes

5. Review Comments to the Author

Reviewer #1: In this study, the authors investigate the phenotypic effects of a brief exposure (2h) to cold and hot temperatures in honeybee queens. The authors did not detect any effect of the temperature treatments on queen mass and egg production, worker mass, sperm viability and embryo protein content. The manuscript is centered around an applied framework, raising the question whether exposure of queens to fluctuating temperatures during the commercial shipping of honeybees affects queen performance. In my opinion, this study also has a broader interest for any biologists interested in the phenotypic effects of temperature variation.

The study is well designed, the manuscript is well written, and the methods and results are clearly presented. I only have minor comments (listed below), but otherwise recommend the publication of this manuscript in Plos One.

L60: Should be “Aphidius”.

L87: I find the “shipping angle” of the manuscript a bit reductive, and in my opinion, the interest of the study goes beyond this applied perspective.

L89: All statistics output are provided as tables. This is fine with me, but I was somehow missing this information in the text.

L94: Here it sounds like the authors are merely repeating an experiment that was previously done – and published. While I find the possible explanations for the discrepant results very convincing and well discussed, I kept wondering why the authors repeated this (part of the) study.

L178: This has really never been investigated? Even transgenerational effects of pollutant exposure (for example)?

L204: Could this be linked to their mite-free hives?

L412: This is usually not recommended to include a random variable with less than 5 levels (here, 2 levels). This could result in a bad estimate of the variance explained the random effect, and could alter the ability of the model to detect fixed effects (for example via an overestimated variance for the random effect, resulting in an underestimated variance to be potentially explained by the fixed effects). Because the authors report “negative results” (which in my opinion is not an issue per se), I am wondering whether the decision to include queen source as random variable affected the results. Another comment is that the manuscript would benefit from reporting power analyses to support the claim of the authors that there is no effect of temperature (and that the study had enough power to detect one if there was one).

Romain Libbrecht

Reviewer #2: Dear authors,

I think it is interesting data, but I am puzzled by the motivation for the study.

The main motivation is potential effects during transport, but the study does not control for transport at all. The spermatheca has evolved to keep the sperm viable, the ability of a colony to maintain temperature and humidity ensures the right conditions, so the only thing can happen during the transport. I would expected a more evolutionary approach like the haploidy hypothesis, also I disagree with the conclusion that it is surprising that queens don’t show the same susceptibility like the drones. The queens has to fly out one to three times, she does the heavy lifting during the mating and she has to return to the colony. The drone might has to fly out, perhaps several times before he is successful. But he has not to find the same colony again.

What I want to say is to include more of the biology of the system to develop the research question.

I am not sure that the group “hive” is a proper control in that case. Since the queen is exposed to varying temperatures during the mating flight or during swarming.

So you have cold, hot and normal as a treatment, but not a control in the strict sense and I am not sure that you need a control or how that would like like in this setup.

Also I would have expected to see any kind of power analysis to give an indication of the validity of the results. The simplest explanation for the non-significant results is that the sample size is too small.

Why did you uses ratios instead of including repeated measurement in your analysis?

I think the section on the potential venom contamination is quite comprehensive, however would it be possible to confirm with another proteomics data set which is NOT from your lab to be able to exclude a “lab effect”

The figures are rather small in my opinion, perhaps one could increase the size. Also, ISO units should be used - not sure what 1/2 lb is.

6. PLOS authors have the option to publish the peer review history of their article (what does this mean?). If published, this will include your full peer review and any attached files.

Reviewer #1: **Yes: **Romain Libbrecht

Reviewer #2: No

---

## [Author Response · Author response to Decision Letter 0]

1 Jul 2021

Point by point response to reviewers

Reviewer #1: In this study, the authors investigate the phenotypic effects of a brief exposure (2h) to cold and hot temperatures in honeybee queens. The authors did not detect any effect of the temperature treatments on queen mass and egg production, worker mass, sperm viability and embryo protein content. The manuscript is centered around an applied framework, raising the question whether exposure of queens to fluctuating temperatures during the commercial shipping of honeybees affects queen performance. In my opinion, this study also has a broader interest for any biologists interested in the phenotypic effects of temperature variation. The study is well designed, the manuscript is well written, and the methods and results are clearly presented. I only have minor comments (listed below), but otherwise recommend the publication of this manuscript in Plos One.

Thank you for the thoughtful review and guidance for improving the manuscript. We have addressed each of your concerns to the best of our ability below.

1. L60: Should be “Aphidius”.

Done. 

2. L87: I find the “shipping angle” of the manuscript a bit reductive, and in my opinion, the interest of the study goes beyond this applied perspective.

The interest of the study may indeed go beyond an application to queen shipping; however, we and others have not yet actually shown that queens can be exposed to damaging temperatures when inside their hives. We have found that temperatures inside hives can vary greatly, spiking up to 40 C on the outer frames during heat waves (McAfee et al. Nature Sustainability), but the queens may very well avoid those hotter areas and stay in the cooler core. Given that the biological relevance to adult queen temperature stress is not certain, we feel the shipping angle is a more solid justification for the research. We have added a sentence in the introduction explaining that risk of temperature stress inside hives is not known for queens (Line 41):

“While some data suggests that temperature stress inside hives is theoretically possible [8, 12], and extreme ambient temperatures are associated with colony losses [13], the actual risk that in-hive temperature fluctuations pose to queens is not known. This is because the core of colony is remarkably thermostable, whereas the periphery is the variable zone [8, 12] – queens could therefore avoid temperature stress by remaining in the center of the brood nest.”

3. L89: All statistics output are provided as tables. This is fine with me, but I was somehow missing this information in the text.

We referenced Table 1 at the first mention of specific results, but it was in parentheses and easy to miss. We instead referenced the table in a complete sentence after the first reporting of results (line 98):

“Statistical parameters, including the test used, degrees of freedom, t or z statistics, and p values, are summarized in Table 1.”

4. L94: Here it sounds like the authors are merely repeating an experiment that was previously done – and published. While I find the possible explanations for the discrepant results very convincing and well discussed, I kept wondering why the authors repeated this (part of the) study.

We feel it is good scientific practice to confirm expected results, even if demonstrated numerous times before. Since the queens were the same ones that were participating in the field trial, additional samples did not even need to be generated (it was not a new experiment). The actual test of sperm viability is also quite simple and quick, so it would seem negligent for us not to check how our temperature exposures impacted the queens’ sperm! Finally, operator bias has been documented for sperm viability measurements specifically (Eckel et al., Frontiers in Ecology and Evolution), and to our knowledge, none of the previous publications conducted this analysis blindly, leaving an open question as to bias in those results, and we had an opportunity to reconcile this.

5. L178: This has really never been investigated? Even transgenerational effects of pollutant exposure (for example)?

There is one example (Preston et al.) that we discussed in the introduction but neglected to reference here (that is fixed now, and we expanded the discussion around it in the introduction at line 59), but other than that, yes. It is surprising to us as well, but is driven by the fact that very little research into abiotic stressors is done on queens in general. More typically, either workers or whole colonies are exposed to the stress treatment. In the case of whole colonies, effects of exposure on brood production and subsequent worker or drone physiology have been explored, but since the whole colony was exposed and not the queen specifically, and exposed workers play a critical role in brood care, these results are not attributable to transgenerational impacts specifically. 

“While this is known to directly impact queen fertility via reductions in sperm viability, maternal effects of stress and vertical impacts on progeny are poorly characterized. In one study by Preston et al., the researchers investigated vertical effects of cold stress (4 °C, 2 h) applied to adult queens, and found that cold stress delayed development of embryos and adult emergence, but not adult immunocompetence nor behaviors [15]. Heat stress has not been investigated in this regard, and effects of temperature stress on egg laying pattern, queen mass, and vertical effects on global protein expression profiles have not been investigated.”

Error fixed at line 201: 

“Transgenerational effects of abiotic stressors have been documented in other insects, including hymenoptera [16, 17, 19], but they have only been briefly investigated in honey bees with regard to cold stress, and not heat stress [15].”

6. L204: Could this be linked to their mite-free hives?

This is an interesting point – the robust stress tolerance is not likely linked to using mite-free bee packages, since the queen is not normally parasitized by the mite, but whether or not other stressors might interact with temperature stress in queens is not known. We added a note about this at the end of the paragraph in question (line 233):

“It is also possible that the effect of temperature stress is impacted by extraneous co-exposures to other variables, like pathogens, nutritional stress, or pesticides, which may not always be accounted for.”

7. L412: This is usually not recommended to include a random variable with less than 5 levels (here, 2 levels). This could result in a bad estimate of the variance explained the random effect, and could alter the ability of the model to detect fixed effects (for example via an overestimated variance for the random effect, resulting in an underestimated variance to be potentially explained by the fixed effects). Because the authors report “negative results” (which in my opinion is not an issue per se), I am wondering whether the decision to include queen source as random variable affected the results. Another comment is that the manuscript would benefit from reporting power analyses to support the claim of the authors that there is no effect of temperature (and that the study had enough power to detect one if there was one).

Considering the influence of the random effect is an excellent point and not one we had previously considered. We were not aware of the potential pitfalls with including a random effect with < 5 levels. We have redone the analyses with the random effect removed and found that indeed there is a marginally significant effect of cold stress on sperm viability (p = 0.048) but all other comparisons remain non-significant. We have updated all elements of the manuscript accordingly. Importantly, our discussion around the variable response to temperature stress, in terms of sperm viability, remains relevant, as we still did not detect an effect of heat. 

Regarding the power analysis, we did consider sample size as a possible explanation for the null results; however, we did not pursue this because, looking at figure 1, you can see that laying pattern, mass ratio, and average callow worker mass ratio are all very close to 1.0 (no change) and do not show a discernable trend in one direction or another, indicating a very small effect size. A power analysis would be quite useful if we observed a trend which was not quite significant. However, given that the effect sizes are generally quite low (with the exception of a couple contrasts of interest), that does not seem to be a problem here and we reason that a power analysis has little utility in this situation. We have, however, added a formal analysis in this regard as a new paragraph (line 104), and a statement that for those contrasts which do have appreciable effect sizes, a lack of sufficient sample size may explain those specific null results. We have also added effect sizes to Table 1.

“The effect sizes (Cohen’s d) for all contrasts were low, ranging from d = 0.03 (laying pattern ratios of cold-treated queens compared to controls) to d = 0.62 (the queen mass ratios of cold-treated queens compared to controls) (Table 1), indicating that, assuming a difference between groups does exist, a sample size of n = 33 to 13,740 would be needed in each group to achieve a power of at least 0.80 for all parameters. While a sample size of 33 would be feasible, indicating that perhaps some differences were missed due to insufficient sample sizes, for many of the parameters measured, the corresponding sample size needed is unreasonably high. As a reductionist approach, the average effect size across all our comparisons of interest was 0.34, which would necessitate a sample size of 108 colonies in each group (324 in total), which is not a realistic scale. Rather, the typically small effect sizes indicate that the strength of phenotypic responses to temperature stress is generally low.”

Reviewer #2: Dear authors,

1. I think it is interesting data, but I am puzzled by the motivation for the study.

The main motivation is potential effects during transport, but the study does not control for transport at all. The spermatheca has evolved to keep the sperm viable, the ability of a colony to maintain temperature and humidity ensures the right conditions, so the only thing can happen during the transport. 

Thank you for the constructive criticism of our work. We are not entirely sure what the reviewer is referring to regarding not controlling for transport, though? We reared these queens ourselves (the same yard in which the experiment was done) in order to avoid any confounding effects of temperature variation during transport on commercial international shipments or even from domestic suppliers. Our motivation to explore effects of temperature stress is driven by previously published temperature tracking data of commercial shipments, which show that low and high temperature spikes occasionally occur, and which can damage the queen’s fertility. In this study, we investigate impacts of that kind of temperature spike in a more controlled setting, to further evaluate biological effects of these stressors. 

2. I would expected a more evolutionary approach like the haploidy hypothesis, also I disagree with the conclusion that it is surprising that queens don’t show the same susceptibility like the drones. The queens has to fly out one to three times, she does the heavy lifting during the mating and she has to return to the colony. The drone might has to fly out, perhaps several times before he is successful. But he has not to find the same colony again. What I want to say is to include more of the biology of the system to develop the research question.

This is a difficult comment for us to respond to, as our research question is rooted in our fundamental motivation for the study. We are motivated by a practical, applied research question: Temperature stress occurs during queen shipments, and we want to know if similar temperature stress events can cause adverse impacts on several queen quality and progeny metrics. We do offer an evolutionary context to temperature tolerance later in the manuscript, as it may be relevant to relate it to other research, but this was not what motivated the study. We do not believe that the motivation behind the research should have a bearing on peer review, and defer to the editor regarding how to proceed.

We would like to clarify, though, our comment on drone versus queen susceptibility. We are absolutely not surprised that queens are more tolerant to temperature stress than drones – the haploid susceptibility hypothesis indicates that the diploid females are entirely expected to be more tolerant to stressors in general. Indeed, we do not actually state anywhere that it is surprising that queens don’t show the same susceptibility as the drones. What we say is that, given the extreme susceptibility of drones, the haploid susceptibility hypothesis alone cannot explain the observed magnitude of temperature tolerance differences between the sexes. If the difference between sexes was purely due to drones having a lack of compensatory alleles, we would expect at least some queens to be homozygous for the presumably recessive, deleterious alleles contributing to temperature sensitivity. Yet queens appear to be universally temperature tolerant, in terms of survival. This is all discussed on Line 255-262. We are not entirely sure how to edit this section to be clearer, but we are happy to remove the discussion altogether if it is causing confusion, as it is not central to our conclusions in the rest of the paper.

3. I am not sure that the group “hive” is a proper control in that case. Since the queen is exposed to varying temperatures during the mating flight or during swarming.

So you have cold, hot and normal as a treatment, but not a control in the strict sense and I am not sure that you need a control or how that would like like in this setup.

The “hive” group controls for the act of handling, being placed in an incubator, and transport to and from the laboratory, but does not include the extreme temperature treatment present in the other groups. We think this control is both correct and necessary for this experiment. The queen may be exposed to variable temperatures during natural flights, but we do not expect that such fluctuations would deviate appreciably from our “hive” control. Mating flights typically occur during fair weather (20-30 C) and are relatively short in duration (around 30 minutes, depending how far it is to the congregation area). The queens were at 20-22 C during vehicle transport to the lab, which falls within this range. 

We suppose that a control in the strictest sense would be queens that did not experience transport of any kind, and remained in their hive for the duration of the experiment. This might be an interesting test for future experiments, but it is really asking what is the effect of transport rather than what is the effect of temperature, which is a different question than we are motivated by here. 

4. Also I would have expected to see any kind of power analysis to give an indication of the validity of the results. The simplest explanation for the non-significant results is that the sample size is too small.

Thank you for this suggestion – it was also brought up by the other reviewer and we have now added a paragraph addressing this (line 104) and have added a column with effect sizes to Table 1: 

“The effect sizes (Cohen’s d) for all contrasts were low, ranging from d = 0.03 (laying pattern ratios of cold-treated queens compared to controls) to d = 0.62 (the queen mass ratios of cold-treated queens compared to controls) (Table 1), indicating that, assuming a difference between groups does exist, a sample size of n = 33 to 13,740 would be needed in each group to achieve a power of at least 0.80 for all parameters. While a sample size of 33 would be feasible, indicating that perhaps some differences were missed due to insufficient sample sizes, for many of the parameters measured, the corresponding sample size needed is unreasonably high. As a reductionist approach, the average effect size across all our comparisons of interest was 0.34, which would necessitate a sample size of 108 colonies in each group (324 in total), which is not a realistic scale. Rather, the typically small effect sizes indicate that the strength of phenotypic responses to temperature stress is generally low.”

5. Why did you uses ratios instead of including repeated measurement in your analysis?

We chose to use ratios because of some general limitations of repeated measures analyses – these types of analyses are generally not recommended when the measurement is taken fewer than three times, or when comparing fewer than three groups. In our experimental design, we repeat the measurement only once (before and after exposure) and are only making two group comparisons (cold to control, or heat to control). Therefore, the ratio approach appears to be more appropriate for this analysis. The ratio represents the change in the metric relative to baseline (and therefore is normalized against individual variation, which is a desirable feature). 

6. I think the section on the potential venom contamination is quite comprehensive, however would it be possible to confirm with another proteomics data set which is NOT from your lab to be able to exclude a “lab effect”

This is a very good suggestion. There are very few labs conducting this type of proteomics research on honey bee queens, with the only other one apparently being Boris Baer’s group. They have published work on the spermathecal fluid proteome but that was with older instrumentation (conducted in 2009) and therefore has limited proteome coverage. They only identified 122 proteins in the spermathecal fluid; nevertheless, melittin was actually among them, meaning that not only is it present, but it is apparently one of the most abundant and robustly expressed proteins. We already discuss the presence of putative venom proteins in the paper recently published by Julianna Rangel, which used transcriptomics and not proteomics, but lends evidence along the same vein. We added reference to this paper at Line 141:

“However, melittin transcripts have been previously identified in the spermathecae of mated queens by an independent research group [23] and another group found melittin proteins in both virgin and mated queen spermathecal fluid [24].”

7. The figures are rather small in my opinion, perhaps one could increase the size. 

We expect the reviewer is referring mainly to figure 1, as this figure has boxplots that are quite small. They are so small because we begin the y axis at zero, rather than a higher number that would enlarge the fraction of the axis with plotted data. We specifically chose to plot the data this way, though, because otherwise proportional comparisons that we naturally make when visually evaluating such data are misleading (if the y axis begins at a number higher than 0, the visual proportional change comparing two plots appears larger than it really is). We have not changed the y axis for this reason, but we have added scattered data points over top of the boxplots so that the data distribution can be gauged. We hope this helps.

8. Also, ISO units should be used - not sure what 1/2 lb is.

We have changed this detail to indicate that ½ lb is approximately 227 g at line 352.

---

## [Editor Report · Decision Letter 1]

15 Jul 2021

Queen honey bees exhibit variable resilience to temperature stress

PONE-D-21-11704R1

Dear Dr. McAfee,

We’re pleased to inform you that your manuscript has been judged scientifically suitable for publication and will be formally accepted for publication once it meets all outstanding technical requirements.

Kind regards,

Nicolas Chaline

Academic Editor

PLOS ONE
---

## [Editor Report · Acceptance letter]

19 Jul 2021

PONE-D-21-11704R1 

Queen honey bees exhibit variable resilience to temperature stress 

Dear Dr. McAfee:

I'm pleased to inform you that your manuscript has been deemed suitable for publication in PLOS ONE. Congratulations! Your manuscript is now with our production department. 

Kind regards, 

on behalf of

Professor Nicolas Chaline 

Academic Editor

PLOS ONE